# Chemical Composition of Essential Oil from Flower Heads of *Arnica Chamissonis* Less. under a Nitrogen Impact

**DOI:** 10.3390/molecules24244454

**Published:** 2019-12-05

**Authors:** Danuta Sugier, Katarzyna Olesińska, Piotr Sugier, Małgorzata Wójcik

**Affiliations:** 1Department of Industrial and Medicinal Plants, University of Life Sciences in Lublin, 15 Akademicka Street, 20-950 Lublin, Poland; danuta.sugier@up.lublin.pl (D.S.); katarzynaolesinska@tlen.pl (K.O.); 2Department of Botany, Mycology and Ecology, Institute of Biological Sciences, Maria Curie-Skłodowska University, 19 Akademicka Street, 20-033 Lublin, Poland; piotr.sugier@poczta.umcs.lublin.pl; 3Department of Plant Physiology and Biophysics, Institute of Biological Sciences, Maria Curie-Skłodowska University, 19 Akademicka Street, 20-033 Lublin, Poland

**Keywords:** *Arnica chamissonis* Less., flower heads, essential oil composition, alpha-pinene, p-cymene, germacrene D

## Abstract

Chamisso arnica (*Arnica chamissonis* Less.) is a valuable plant species used in the pharmaceutical industry due to the content of many pharmacologically active substances and the similarity of its chemical composition to that of *Arnica montana*—a medicinal plant commonly used in pharmacy and cosmetics. The similarity of the two plant species implies that chamisso arnica can be a pharmaceutical substitute for the mountain arnica, i.e., an endangered and endemic plant species in Europe. Chamisso arnica extracts exhibit anti-inflammatory and antiradical activity and possesses high antioxidant properties that might be helpful in preventing or delaying the progress of free radical dependent diseases. The attributes of *A. chamissonis* are mainly related to the content and chemical composition of essential oil. Therefore, the objective of this study was to characterize the chemical composition of essential oil derived from *A. chamissonis* flower heads under a nitrogen impact. The experiment was performed on experimental fields in mid-eastern Poland on two soil types (sandy and loamy soils). The nitrogen fertilizer was applied as ammonium sulfate (control, 30, 60, 90, and 120 kg N ha^−1^). Collection of flower heads was carried out in the full flowering phase, which was characterized by the highest content of essential oil. The chemical composition of essential oil was examined using GC-MS. Among the 75 ingredients of the volatile oil of chamisso arnica flower heads, alpha-pinene, cumene, p-cymene, germacrene D, spathulenol, decanal, caryophyllene oxide, beta-pinene, and benzene acetaldehyde were present at relatively high levels. Both the nitrogen application and the soil type had an effect on the oil concentration and the yield of the main constituents (alpha-pinene and germacrene D) with pharmacological value. Different levels of nitrogen application could be considered as a relevant way to modify the chemical composition and to increase the essential oil production.

## 1. Introduction

Essential oils (EO) are important secondary metabolites of plants. They have been used for centuries not only in different fields/branches of industry but also in ethnobotanical medicine [1]. The quantitative and qualitative composition of these compounds is determined by different factors, e.g., plant origin and habitat [2,3], cultivar [4,5], plant development phase [4,6,7], seasonal variations [8,9], weather and climatic conditions [10,11,12], water conditions [13,14], and plant nutrition [7,15,16,17]. Among the essential nutrients, nitrogen is one of the basic elements in the plant structure. It is involved directly or indirectly in the production of secondary metabolites, including EO [16,18]. Primarily, this macroelement has a significant impact on plant biomass and the raw material yields [4,10,19]. In the case of medicinal and aromatic plants, it can also influence the content and chemical composition of EO as well as the EO yield [9]. Its influence depends on many factors, including the response to the environmental and agrotechnical conditions. In literature, there are many examples of different responses of medicinal and aromatic plant species to nitrogen nutrition. Abbaszadeh et al. [15] showed a positive effect of a nitrogenous fertilizer on the biological yield and the content and yield of EO in *Melissa officinalis* L. plants. Other authors reported that the nitrogen application contributed to an increase in the leaf fresh weight and the EO content in *Mentha spicata* L. [20], the dry and fresh mass and the EO content in *Mentha piperita* L. [21], as well as the herb and oil yield in *Satureja hortensis* L. and *Chamomilla recutita* L. Rauschert [5,22]. On the other hand, Karamos and Sotiropoulou [10] showed a positive impact of nitrogen fertilization on the raw material yield and chemical composition but not on the content of EO in *Origanum vulgare* ssp. *hirtum* (Link) Ietswaart. Similarly, Bufalo et al. [19] did not show an impact of this nutrient on the EO concentration in *Ocimum basilicum* L., and Nurzyńska-Wierdak [4] found that the application of nitrogen generally contributed to a decrease in the content of the main components of basil volatile oil: geraniol, 1,8-cineole, eugenol, and germacrene D, but increased the linalool content. However, opposite results were obtained by Sifola and Barbieri [23], who demonstrated that nitrogen fertilization increased the fresh weight yield of herb and leaves as well as the concentration and yield of EO in basil. In turn, Omer [24] showed an increase in the EO content in *Origanum syriacum* L. var. *aegyptiacum* Tackh and demonstrated a decrease in the biosynthesis of a-terpinene and p-cymene in contrast to thymol and carvacrol. Frequently, an increase in nitrogen doses causes an increase in the EO yield due to the higher flower yield in the absence of an influence of this nutrient on the EO concentration in tissues, as demonstrated in the case of *Matricaria recutita* L. [25].

Chamisso arnica (*Arnica chamissonis* Less.) is an herbaceous perennial herb and a medicinal plant that is widely used as an herbal remedy. Its occurrence range extends from the Alaskan Archipelago south to the San Bernardino Mountains in California and from the southern Rocky Mountains to New Mexico [26]. This species is a valuable source of bioactive compounds and herbal raw material [27,28,29]. Extracts from chamisso arnica exhibit anti-inflammatory and antiradical activity and possess high antioxidant capacity that might be helpful in preventing or delaying the progress of free radical-dependent diseases [6,28,30,31]. The properties of *A. chamissonis* are mainly related to the content and chemical composition of EO. Given its content of numerous pharmacologically active substances [32,33], this pharmaceutically valuable plant species is characterized by a similar chemical composition and similar pharmacological effects to those of *Arnica montana*, i.e., a medicinal plant rich in secondary metabolites [6,17] and commonly used in pharmacy, homeopathy, and cosmetics [34,35]. Moreover, in their study of the two arnica species, Gawlik-Dziki et al. [28] showed that free radicals were scavenged more effectively by extracts of *A. chamissonis* seeds than extracts of *A. montana* seeds. Either alone or in combination with other active compounds, the former can be used in cosmetic, nutraceutical, and pharmaceutical applications. It should also be stressed that *A. montana* does not show yield stability in the conventional European agricultural conditions, as shown by field trials conducted in France, Germany, and Switzerland, in contrast to *A. chamissonis* [30,36], which is important for production of the raw material and is accepted in some pharmacopoeias [29,35]. The similar phytochemical profile of the two plant species makes *A. chamissonis* a suitable species to substitute *A. montana* for pharmaceutical purposes [28,30,32], especially considering that the latter is critically endangered in most European countries [37]. Therefore, investigation of factors modifying the chemical composition of *A. chamissonis* raw material and introduction of this plant species to cultivation tests on different soil types can help to reduce the pressure on the endangered and highly valued medicinal *A. montana* [38] and limit uncontrolled harvesting of inflorescences, which has a negative effect on the native populations of this species.

Studies have indicated that the dominant EO components in the flower heads of *A. chamissonis* are similar to those found in *A. montana* [6,17,39], including molecules with significance emphasized and confirmed in the literature. The main EO component of *A. chamissonis* is alpha-pinene exhibiting antioxidative, anti-inflammatory, antimicrobial, and antiparasitic bioactivity [17,40,41,42,43] as well as a significant effect on the inhibition of tumor invasion [44]. Another important component, caryophyllene oxide, is a dominant component in the EO of some Asteraceae species [45,46,47]. It is characterized by biological activities, such as anti-inflammatory, antibacterial, antiparasitic, and insecticidal effects [47,48,49,50]. Moreover, EO containing caryophyllene oxide exerts an allelopathic effect on the germination and growth of weed seedlings [51]. In turn, volatile oils containing germacrene D exhibit anticancer, antifungal, and antibacterial activity [52,53,54,55]. Our former study [17], which was the first comprehensive study on the effect of boron application on the chemical composition of arnica EO, showed an increased yield of alpha-pinene, caryophyllene oxide, and germacrene D but a reduced yield of p-cymene and decanal in EO of *A. chamissonis* flower heads following boron treatment. However, an increased yield of caryophyllene oxide, germacrene D, and decanal and a reduced yield of carvacrol and n-hexadecane were found in the EO of *A. montana* [17]. Therefore, in studies on the influence of nutrients on the chemical profile of EO, nitrogen emerges as one of the most interesting and promising factors for acquisition of a new combination of molecules in *A. chamissonis* essential oil to be used in the pharmaceutical industry.

While analyzing the aforementioned examples of the role of nitrogen in crop generation and modification of the chemical composition of EO in medicinal and aromatic plant species, it can be expected that the modification of the nitrogen level may result in a higher EO yield and new EO quality in the case of *A. chamissonis* as well. Since the interest in medicinal plants is continuously growing due to the increasing consumer demand [56], a new combination of EO compounds may ensure new properties and biological activities that can be used in medicine, pharmacy, or cosmetic industry. In the case of *Arnica* spp., the knowledge of the role of nutrients as factors modifying the content, yield, and chemical composition of EO is insufficient [17,29]; therefore, the results presented in this paper partially fill in this gap. Another argument confirming the validity of our research is the rapid growth in the industrial demand for medicinal plants and herbal medicines worldwide, which has contributed to intensification of the cultivation of the species and search for new sources of secondary metabolites in recent years [17,31,57,58,59,60]. The effects of application of nutrients or the soil type and properties on the quality and quantity of secondary metabolites in EO from arnica flower heads have not been investigated so far. Therefore, the objectives of this study were: (i) to characterize the chemical composition of EO derived from *A. chamissonis* cultivated in the field conditions of eastern Poland; (ii) to evaluate the influence of different nitrogen doses and soil type on the quantity, chemical composition, and yield of EO in the flower heads of *A. chamissonis*.

## 2. Results

### 2.1. Content and Yield of Essential Oils

The content of EO in the flower heads of *A. chamissonis* and the EO yield under nitrogen fertilization applied into the two different soil types are presented in Table 1. There was a positive and statistically significant effect of nitrogen fertilization, soil type, and their interaction on the crop yield, EO content, and EO yield in the flower heads of the studied plant species. It was found that the increase in the N rate from 0 to 120 kg ha^−1^ resulted in an increased crop yield of *A. chamissonis* flower heads from 1210 to 1941 kg ha^−1^ and from 994 to 1563 kg ha^−1^ on the loamy (L) and sandy (S) soils, respectively.

The increase in the N rate from 0 to 90 kg ha^−1^ also resulted in enhanced accumulation of EOs in the *A. chamissonis* flower heads (from 0.151 to 0.180% and from 0.137 to 0.162% on the L and S soils, respectively). A further increase in the N dose up to 120 kg ha^−1^ had no effect on the content of volatile oils in the *A. chamissonis* flower heads in any soil type. The N fertilization differentiated the EO yield significantly (Table 1). An increase in the EO yield was observed after the successive increase in the N dose on both the L and S soils (from 1829.6 to 3488.8 g ha^−1^ and from 1354.9 to 2529.1 g ha^−1^, respectively), i.e., by ca. 90% and by ca. 86%, respectively. The EO content and yield were higher in the raw material taken from plants growing on the L soil than on the S soil at each fertilization level. Moreover, the level of the essential oil yield in *A. chamissonis* flower heads cultivated without fertilization on the L soil was comparable with that on the S soil at the 30 kg ha^−1^ N rate.

### 2.2. Chemical Composition and Diversity of Volatile Oils

Alpha-pinene, cumene, p-cymene, germacrene D, spathulenol, decanal, caryophyllene oxide, beta-pinene, and benzene acetaldehyde were the main components of the EO from the *A. chamissonis* flower heads (Table 2, Figure 1 and Figure 2). There were significant qualitative and quantitative differences in the chemical composition of the EO in plants exposed to the N fertilization on the two soil types. Nitrogen fertilization modified especially the number of EO components in plants grown on the S soil. In turn, the number of EO components obtained from plants grown on the L soil increased from 65 to 71 upon the N fertilization. The essential oil compounds constituted from 93.91 to 97.74% in the raw material taken from plants grown on the S soil and from 96.33 to 99.96% in the raw material collected from plants grown on the L soil.

### 2.3. Differentiation of the EO Content

The PCA of all the 75 chemical variables from the studied samples revealed two principal components with a higher influence on the EO chemical composition, but only five main components are presented in Figure 1. The analysis showed that the two separate groups, i.e., samples S0–120 and L0–120, were clearly distinguished among the samples studied. The two PCA axes account for 85.72% of the total variance, with 70.32% of the total variance explained by the first axis; therefore, two principal components are sufficient to describe the presented samples (Table 3). Axis 1 is the linear combination of the studied oils, which better summarizes the variations in the original data matrix in a single number, whereas Axis 2 better summarizes the remaining information. The dimensionality of the data was therefore reduced from 75 variables to two uncorrelated components with 5% loss of variation.

The chemical variables are represented as a function of both Axis 1 and Axis 2. Alpha-pinene, germacrene D, p-cymene, cumene, and spathulenol are the main factors determining the chemical differentiation of EOs (Figure 3, Table 3). Axis 1 shows a high positive correlation with alpha-pinene and a negative correlation with spathulenol; therefore, this principal component separates samples with high content of alpha-pinene and low content of spathulenol (L0, L30, L60, L90, L120) and samples with low content of alpha-pinene and high content of spathulenol (S0, S30, S60, S90, S120). In turn, Axis 2 of PCA shows a high positive correlation with germacrene D, cumene, and spathulenol and a negative correlation with p-cymene. This principal component separates samples situated in the upper part of the ordination space with high content of germacrene D, cumene, and spathulenol and samples located in the lower part of the ordination space with high content of p-cymene.

### 2.4. Yield and Diversity of Main EO Components

Among the nine main components of the volatile oil from the *A. chamissonis* flower heads, whose content exceeds 3% (Table 2), alpha-pinene, cumene, p-cymene, germacrene D, spathulenol, decanal, caryophyllene oxide, beta-pinene, and benzene acetaldehyde are characterized by a relatively high yield (Table 4). There was a positive and statistically significant effect of N fertilization, soil type, and their interaction on the yield of the main EO components. It is worth emphasizing that, already in the control sample (0 kg N ha^−1^), an over 2-fold higher yield of alpha-pinene, cumene, germacrene D, beta-pinene, and p-cymene and over 12-fold higher amounts of benzene were recorded in the case of flower heads of arnica plants growing on the L soil in comparison to the plants from S soil. In contrast, the spathulenol yield was higher in plants growing on the S soil than on the L soil. The yield of decanal and caryophyllene oxide in the control samples was similar in the raw material obtained from the two soil types. In the case of plants cultivated on both L and S soils, the increase in the N dose caused an increase in the yield of the EO components.

The increase in the N rate from 0 to 120 kg ha^−1^ on the S soil resulted in a higher yield of alpha-pinene (from 196.82 to 542.30 g ha^−1^), cumene (from 149.37 to 252.92 g ha^−1^), germacrene D (from 74.55 to 153.68 g ha^−1^), spathulenol (from 101.23 to 129.13 g ha^−1^), decanal (from 85.27 to 112.66 g ha^−1^), caryophyllene oxide (from 77.57 to 126.84 g ha^−1^), and beta-pinene (from 30.53 to 78.23 g ha^−1^). An increase in the N dose to 60 kg ha^−1^ on this soil type resulted in an increase in the p-cymene yield from 70.83 to 183.20 g ha^−1^; however, a further increase in the N dose caused a statistically significant decrease in this EO component. In the case of benzene acetaldehyde, the yield increased with the increase in the N dose to 90 kg ha^−1^, whereas the increase in the N dose to 120 kg ha^−1^ did not cause a significant increase in the yield.

It was found that the increase in the N rate on the L soil from 0 to 120 kg ha^−1^ resulted in a growing yield of cumene (from 205.52 to 366.10 g ha^−1^), p-cymene (from 134.83 to 273,19 g ha^−1^), beta-pinene (from 65.96 to 122.15 g ha^−1^), and benzene acetaldehyde (from 62.49 to 147.21 g ha^−1^). In the case of alpha-pinene, the N dose 90 kg ha^−1^ caused an increase in the yield from 482.65 to 894.17 g ha^−1^, whereas the application of 120 kg N ha^−1^ did not cause a further increase in the yield. The yield of the other main components of the volatile oil of *A. chamissonis* growing on the L soil also increased under the N fertilization: germacrene D (from 90.74 to 161.82 g ha^−1^), spathulenol (from 73.79 to 139.90 g ha^−1^), decanal (from 81.63 to 130.85 g ha^−1^), and caryophyllene oxide (from 72.52 to 154.86 g ha^−1^). Of note, the N dose of 120 kg ha^−1^ caused a statistically significant decrease in these EO components (Table 4).

Figure 4 presents the oil yield from *A. chamissonis* distributed in the ordination space of PCA. The two PCA axes account for 97.42% of the total variance, with 94.52% of the total variance explained by the first one; therefore, two principal components are sufficient to describe the studied samples (Table 5). The variables (basic essential oil components) are represented as a function of both Axis 1 and Axis 2. The first axis shows a high positive correlation with the yield of alpha-pinene, germacrene D, p-cymene, benzene acetaldehyde, cumene, caryophyllene oxide, and beta-pinene. This principal component separates samples with a high yield of these compounds on the right side from those on the left side with a low yield of these compounds. Axis 2 of PCA shows a high positive correlation with germacrene D and spathulenol and a negative correlation with p-cymene and benzene acetaldehyde (Table 5). This principal component separates samples with a high yield of germacrene D and spathulenol in the upper part and those with a high yield of p-cymene and benzene acetaldehyde in the lower part of the ordination space. Generally, samples with the highest yield of the main EO components can be distinguished on the right side of the ordination space (L90, L12), whereas the left side shows samples with the lowest yield of the main EO components. However, in the central part, there are samples with similar and medium values of the yield of the components (S60, L0, S90, L30, S120).

## 3. Discussion

### 3.1. Raw Material and the Concentration and Yield of Essential Oils

In recent years, many experiments on the introduction of *Arnica* spp. were carried out in many regions of Europe with a perspective to obtain a promising source of plant substance *Arnicae flos* [6,10,12,17,29,63,64], but very few studies were focused on *A. chamissonis* [6,12,29,63]. The experiments were conducted mainly in mountain regions. In experiments carried out on Tara Mountain, Radanović et al. [64] obtained ca. 600 kg ha^−1^ of dry flower heads of *A. chamissonis*. Other researchers obtained ca. 300 kg ha^−1^ in the mountain region of Croatia [12] and 58 kg ha^−1^ in the mountains in Bulgaria [63]. In the present study, the increase in the N fertilization up to 120 kg N ha^−1^ resulted in an increased crop yield of *A. chamissonis* flower heads on the L and S soils, i.e., from 1210 to 1941 kg ha^−1^ and from 994 to 1563 kg ha^−1^, respectively (Table 1). The differences in the flower head yield between the aforementioned reports and our results are evident, which means that the climatic conditions of mid-eastern Poland are more favorable for *A. chamissonis* cultivation. Evidently, to achieve high yields, arnica should be grown on soils rich in nutrients with granulometric composition of clay.

Similar to *A. montana* [11,17,39], *Arnica chamissonis* is characterized by a high concentration of EOs, which are rich in molecules characterized by high biological activity, in comparison with other plant species from the family Asteraceae [65,66,67]. The concentration of oils in the *A. chamissonis* flower heads obtained from the L and S soils ranged from 0.151% to 0.180% and from 0.137% to 0.162%, respectively. The EO concentration in arnica flower heads is higher than in other plant species from this family, e.g., *Monticalia greenmaniana* [53] and see comment in PubMed Commons below *Leucanthemum vulgare* [46], but some species are characterized by a higher or similar concentration of EOs in dried aerial parts, e.g., *Baccharis antioquensis* and *Diplostephium antioquense* [66]. The EO contents presented in this paper are comparable to the results of our earlier studies on EOs in this species and similar to the EO content in *A. montana* flower heads [6,17]. The EO content is substantially higher in relation to data obtained in Serbia [39], where the authors reported that the content of EOs derived from *A. chamissonis* was approximately 0.08% *v*/*w*.

The content, yield, and composition of EOs can be significantly modified by macro- and microelements [16,17,68,69,70]. In the present study, the N effect on the yield of *A. chamissonis* flower heads, oil concentration, and oil yield was evident in the case of the two soil types; however, these parameters were higher in plants grown on the L soil. From the point of view of pharmaceutically useful secondary metabolites, a very important factor is the quality of soil where plant raw material is produced. Despite the increasing number of experiments on the introduction of chamisso arnica to cultivation in Europe conducted in the last years, the effect of the soil type on successful cultivation has not been tested so far [12,29,63]. The available literature reports diverse responses of medicinal and aromatic plants to N application: (i) a positive impact of N fertilization on the raw material yield and the chemical composition of EO but a negative impact on the EO concentration [4,10,19]; (ii) a positive impact on the chemical composition of EO, raw material yield, and essential oil yield but no influence on the EO concentration [25]; and (iii) a positive influence on the raw material yield, EO content, and essential oil yield [5,15,23,35], which was also confirmed in the present study of *A. chamissonis*. The various factors used in experiments often affected the EO yield but did not affect the EO concentration and chemical composition [69,70,71,72,73,74]. Nitrogen fertilization can also be a promising approach with a positive influence on the oil concentration, oil yield (ca. 90% increase in the EO yield), and chemical composition of EO in *A. chamissonis* flower heads. Moreover, N fertilization results in a substantially higher increase in the EO concentration and EO yield than supplementation with boron. Comparison of the data presented in this paper with the data from our former study of boron fertilization [17] showed that N fertilization caused a ca. 50% increase in the raw material yield, a ca. 20% increase in the EO concentration, and a ca. 20% increase in the EO yield. In turn, the values of the increase in the case of boron addition were 19%, 19%, and 42%, respectively. Therefore, nitrogen seems to be an important factor modifying the production and chemical profile of EO *A. chamissonis*.

### 3.2. Chemical Composition and Diversity of Volatile Oils

Seventy-five compounds constituted from 93.91 to 99.96% of the total EO in the flower heads in *A. chamissonis* cultivated on the L soil at the 90 kg ha^−1^ N dose and on the S soil at the 30 kg ha^−1^ N dose. The number of essential oil ingredients was high in comparison with other common Asteraceae species, such as *Artemisia absinthium* L. [75] or *Achillea millefolium* L. [76]. However, it was distinctly lower than that in *Helichrysum* spp. [77] and *Tanacetum vulgare* L. [70]. The component diversity of the *A. chamissonis* EOs, namely the dominance of alpha-pinene, cumene, decanal, p-cymene, germacrene D, and caryophyllene oxide was similar to that demonstrated by Ristić et al. [39], Kowalski et al. [6], and Sugier et al. [17]. In our former research [17], boron fertilization caused an increase in the concentration of the main EO ingredient, i.e., alpha-pinene (from 8.10 to 19.52%). This was similar to the increase in the content of this compound from 14.11 to 21.68% induced by the N fertilization in the present study. Nitrogen also contributed to an increase in the concentration of p-cymene and a decrease in the amount of decanal, germacrene D, and spathulenol; however, boron caused an increase in the level of caryophyllene oxide and a decrease in p-cymene [17]. It should be mentioned that changes in the particular EO compounds were registered only in plots of the S soil, which is less fertile than the L soil used in the experiment with boron fertilization [17]. On the other hand, the chemical composition of the *A. chamissonis* EO is totally different from the data reported by Roki et al. [29], who investigated plants growing on Tara Mountain (subalpine climate) in Serbia. The essential oil composition was dominated by n-pentacosane (9.5%), n-tricosane (6.5%), n-heptacosane (3.2%), n-tetracosane, caryophyllene oxide (5.5%), and spathulenol (3.6%). It cannot be excluded that climatic conditions are the main determinants of the chemical oil composition. The intraspecific variability of the EO components was noted in some Asteraceae plant species, e.g., *A. millefolium* [76], *A. absinthium* [75], *Artemisia vulgaris* L. [78], or *Helichrysum italicum* m (Roth) G. Don [77]. Similarly, EOs from different European *A. montana* populations have shown chemical diversity [6,39,45,67].

As demonstrated in this paper, the nitrogen application modified the EO composition. However, very interesting is the fact that the proportion (from over a dozen to several dozen percent) of the main EO components from the flower heads of this species, i.e., alpha-pinene, caryophyllene oxide, and germacrene D, is similar to the chemical oil profile in such species as *Centaurea* spp. [79], *Eupatorium intermedium* DC. [47], and *Copaifera langsdorffii* Desf. [80]. Alpha-pinene is the main constituent of *Cupressus sempervirens* L. cone EO and *Pistacia vera* L. [81,82]. In turn, germacrene D, spathulenol, and beta-pinene were the main components of *Santolina africana* Jord. et Fourr. [83]. Alpha-pinene, beta-pinene, and germacrene D were identified as the main components of the EO of *Senecio vernalis* Waldst. & Kit. grown in Turkey [84].

In the present study, monoterpene hydrocarbons were the most abundant group, representing from 47.01 to 53.74% in the EO obtained from plants cultivated on the L soil and from 35.71 to 48.21% in EO obtained from plants cultivated on the S soil. The second largest group comprised sesquiterpene hydrocarbons, which accounted for 10.96–13.87% and 11.65–19.15% in the EO obtained from plants cultivated on the L and S soil, respectively. Additionally, based on the comparison of the control samples from the two soil sites, it can be concluded that L soils, which are rich in macroelements and are more fertile than S soils, favor production of monoterpene hydrocarbons rather than sesquiterpene hydrocarbons. In this case, the habitat characteristics [2,3] and plant nutrition [7,15,16,17] seem to be very important determinants of the EO chemical composition in chamisso arnica. In turn, the comparison of the control samples presented in this study (Table 2) with the control samples in our previous study [17] shows substantial differentiation in the chemical composition of EO. It must be noted that, in both field experiments, the same *A. chamissonis* genotype was cultivated in the same site and on the same sandy soil, and the flower heads were collected in the full flowering phase, which is characterized by the highest content of EO, but in different weather conditions. Probably, this factor, regarded as a determinant of the quantitative and qualitative EO composition [10,11,12], had the greatest influence on the content of alpha-pinene, germacrene D, and p-cymene, in the absence of an influence on decanal and caryophyllene oxide.

### 3.3. Yield of Main Components of Volatile Oils

Among the nine main ingredients of the volatile oil of *A. chamissonis* flower heads with content exceeding 3%, there was a relatively high yield of alpha-pinene, cumene, p-cymene, germacrene D, spathulenol, decanal, caryophyllene oxide, beta-pinene, and benzene acetaldehyde. It is worth emphasizing that, already in the control sample (0 kg N ha^−1^), the yield of alpha-pinene, cumene, germacrene D, beta-pinene, and p-cymene was over 2-fold higher. The yield of benzene acetaldehyde was over 12-fold higher in the EO of flower heads in plants growing on the L soil in relation to the S soil; in contrast, the spathulenol yield was higher in plants growing on the S soil than the L soil. The yield of decanal and caryophyllene oxide in the control samples was similar in the raw material obtained from the two soil types. Both in the cases of plants cultivated on the L soil and on the S soil, the increase in the nitrogen dose caused an increase in the yield of the EO components.

The present study showed that the increase in the N rate from 0 to 120 kg ha^−1^ in the S soil resulted in a higher yield of some components of EO, including alpha-pinene, cumene, germacrene D, spathulenol, decanal, caryophyllene oxide, and beta-pinene. In turn, the increase in the N dose to 60 kg ha^−1^ and 90 kg ha^−1^ in this type soil caused an increase in the p-cymene and benzene acetaldehyde yields, respectively.

The successive N fertilization caused an increase in the alpha-pinene concentration in the EO of *A. chamissonis* flower heads, as in the case of boron fertilization [17]. Therefore, in further studies of factors modifying the EO chemical composition and influencing the yield of the main and very important EO compounds, attention should be paid to the simultaneous use of these two elements and their interaction.

### 3.4. Role and Value of the Main Components of Essential Oils from A. Chamissonis Flower Heads

The compounds present in the *A. chamissonis* EO determine the biological activity of formulations derived from this plant raw material. There are sparse data on *A. chamissonis* volatile oils despite the fact that the EO contributes to all medicinal properties of the plant. Alpha-pinene, germacrene D, p-cymene, and caryophyllene oxide are the main components of essential oils in *Arnica* spp. [6,17]. Alpha-pinene is a monoterpene compound with a wide spectrum of bioactivity, including antioxidative [41], anti-inflammatory [42], antimicrobial [40,85], antiparasitic [43], anticancer [41], and antinociceptive [86] properties. Moreover, this main component of *A. chamissonis* EO is the primary compound present in numerous medicinal plant species belonging to the Asteraceae family, e.g., *Chrysanthemum coronarium* L. [68], *Eupathorium buniifolium* Hook. Ex Hook & Arn. [74], and *Monticalia greenmaniana* (Hieron) C. Jeffrey [53], or representatives of other families, e.g., *Ducrosia anethifolia* Boiss [85]. Alpha-pinene has a significant effect on the inhibition of tumor invasion and may potentially be developed into an antimetastatic drug [43]. Moreover, this oil component is characterized by apoptotic and antimetastatic activity [87]. Alpha-pinene isolated from plants can be a valuable component for the therapy of melanoma, given its great potential to induce apoptosis in cancer cells [87]. The high concentration of this EO component in the *A. chamissonis* raw material and the high yield of this compound, especially under the nitrogen fertilization, make this plant very interesting as a source of important secondary metabolites that can be produced in field conditions, but definitely on very fertile nutrient-rich soils with a granulometric composition of clay. Decanal is another compound exhibiting antibacterial [85,88], antioxidant, and antitumor [88] activities and is a basic component of the *A. chamissonis* oil. It is one of the main components found in some plant species, e.g., *A. montana* [6,17]. Caryophyllene oxide is an oxygenated sesquiterpene present as a dominant component in the EO of some Asteraceae plant species [46,47]. This molecule is characterized by anti-inflammatory and analgesic, antibacterial, antifungal, antiparasitic, and insecticidal activities [47,48,49,50].

Germacrene D is a nonoxygenated sesquiterpene present as a dominant component in the EO of plant species in the Asteraceae family, e.g., *A. millefolium* [76], *A. vulgaris* [89], *E. intermedium* [47], *M. greenmaniana* [53], and *S. africana* [83]. It is worth adding that spathulenol and beta-pinene, detected in the EO of the latter species [83], were also found in the flower heads of chamisso arnica, where they accounted for over a dozen percent. Volatile oils containing germacrene D exhibit anticancer [55], antifungal [54], antibacterial [53], antioxidant, and anti-inflammatory activities [83]. This bioactive molecule was previously detected as a major component in the oil of *Cananga odorata* Hook. F. & Thomson flowers [89,90] with antimicrobial activities. The EO extracted from *C. odorata* flowers is an important raw material for the cosmetic and perfume industry [90]. The high content of this EO ingredient in *A. chamissonis* flower heads, especially under N fertilization, makes the plant species attractive for use in the cosmetic and perfume industry. Given the similarity of the chemical composition of oils characteristic for both arnica species (*A. montana*, *A. chamissonis*), it can be suggested that the analyzed group of secondary metabolites should exhibit common biological properties, and both boron [17] and nitrogen can be important regulators of the EO composition in *A. chamissonis* flower heads and essential oil-generating factors.

## 4. Materials and Methods

### 4.1. Plant Material and Soil Conditions

The *Arnica chamissonis* Less. species used in the experiment was identified by Anna Rysiak, a taxonomist from the Maria Curie-Skłodowska University in Lublin based on the reference material from the collection of medicinal plants of the Department of Industrial and Medicinal Plants of the University of Life Sciences in Lublin. In 2017, the raw material for the chemical analyses was taken from 2-year-old *A. chamissonis* individuals from two experimental fields at the University of Life Sciences in Lublin located in the eastern part of Poland. The first experimental field (51°31′25″N; 22°45′04″E) was located on sandy (S) soil (sand 66.2%, silt 20.5%, clay 13.3%), and the other experimental field (51°29′28″N; 22°51′18″E) was located on loamy (L) soil (sand 21.5%, silt 42.1%, clay 36.4%). The sandy soil was characterized by moderate content of organic matter (1.67%; PB-34—Tiurin method), moderate phosphorus (P_2_O_5_—138 mg kg^−1^; PN-R-04023:1996), low potassium (K_2_O—79 mg kg^−1^; PN-R-04022:1996+Az1:2002), and very low magnesium levels (Mg—13 mg kg^−1^; PN-R-04020:1994+Az1:2004), with very acidic reaction. In turn, the loamy soil was characterized by high content of organic matter (1.88%), very high phosphorus (258 mg kg^−1^), high potassium (39 mg kg^−1^), and moderate magnesium levels (51 mg kg^−1^), with neutral reaction.

### 4.2. Raw Material Collection

*A. chamissonis* flower heads were collected in the full flowering phase. This phase is characterized by the highest content of EO in the flower heads [11]. The flower heads were dried in a drying chamber at 40 °C immediately after the harvest.

### 4.3. Qualitative and Quantitative Analysis of Essential Oil

#### 4.3.1. Assay of the Essential Oil Content

Twenty grams of powdered chamisso arnica inflorescences were submitted to water-distillation in a Deryng apparatus with 500 mL water for 3 h according to the Polish Pharmacopoeia VI [91]. The method of indirect distillation was applied (with xylene). The essential oils were collected over water, separated, dried over anhydrous sodium sulphate, and stored in the dark at 4 °C prior to GC–MS analysis. The analysis was carried out in four repetitions.

#### 4.3.2. GC-MS Analysis

The chromatographic analysis was performed according to procedures described previously [11,17]. The analysis was performed in triplicate. The essential oils were analyzed using a Varian 4000 GC–MS/MS system (Varian, Palo Alto, CA, USA). The compounds were separated on a 30 m × 0.25 mm × 0.25 μm VF–5 ms column (Varian, USA). The column temperature was increased from 50 to 250 °C at a rate of 4 °C/min; injector temperature 250 °C; split ratio 1:50; injection volume 5 μL. The MS parameters were as follows: EI mode, with ionization voltage 70 eV; ion source temperature 200 °C; scan range 40–870 Da.

#### 4.3.3. Qualitative and Quantitative Analysis

The qualitative analysis was carried out on the basis of MS spectra, which were compared with the spectra of the NIST library [92] and with data available in the literature [61,93]. The identity of the compounds was confirmed by their retention indices [61] taken from the literature [62,93] and our data for standards described previously [11,17]. The quantitative analysis was performed based on calibration curves and additionally confirmed by means of internal standard addition method (alkanes C_12_ and C_19_) according to procedures described by Kowalski and Wawrzykowski [94].

### 4.4. Statistical Analysis

The two-way analysis of variance (ANOVA) and subsequent Tukey’s tests were used. The differences were considered significant at *p* < 0.05. The statistical analyses were carried out using the Statistica 6.0 software. Principal component analysis (PCA) was applied in order to explain the relationships between oil ingredients and to show variability factors. Prior to the PCA, the data on the EO content and oil yield of the main components (over 3%) were centered. The analyses were carried out using the statistical package (MVSP) program version 3.1. [95].

## 5. Conclusions

The attributes of *Arnica chamissonis* are mainly related to the content and chemical composition of essential oil. Among the 75 ingredients of the essential oil from *A. chamissonis* flower heads, alpha-pinene, cumene, p-cymene, germacrene D, spathulenol, decanal, caryophyllene oxide, beta-pinene, and benzene acetaldehyde were characterized by relatively high levels. Both the nitrogen application and the soil type, which had not been tested so far, had an effect on the concentration and yield of the main constituents, such as alpha-pinene, decanal, and germacrene D, i.e., molecules of high pharmacological value. Different levels of nitrogen application can be considered as a relevant way to modify the chemical composition and to increase the raw material and essential oil production. In further studies of factors modifying the EO chemical composition and influencing the EO concentration and yield of the main and very important EO compounds, attention should be paid to the simultaneous use of nitrogen and other microelements to achieve synergistic results. The flower head yield, EO concentration, and essential oil yield obtained in the presented study show that the climatic conditions of mid-eastern Poland are favorable for *A. chamissonis* and a wide range of pharmacologically active substances can be produced by this plant species in this region of Europe.

## Figures and Tables

**Figure 1 molecules-24-04454-f001:**
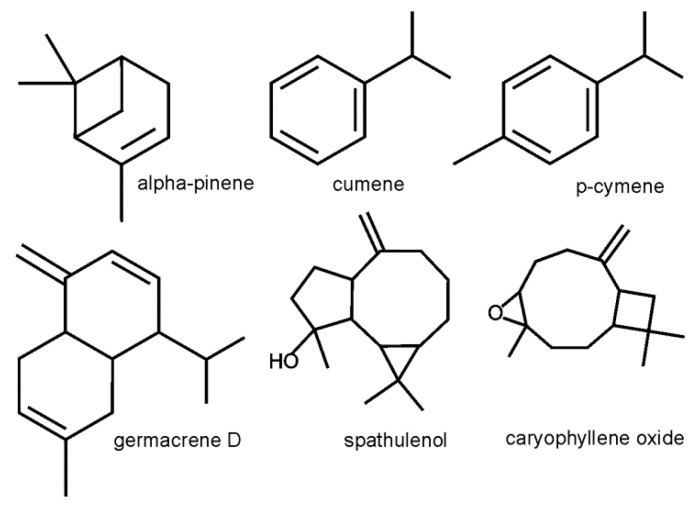
Chemical structures of selected EO compounds in flower heads of *A. chamissonis.*

**Figure 2 molecules-24-04454-f002:**
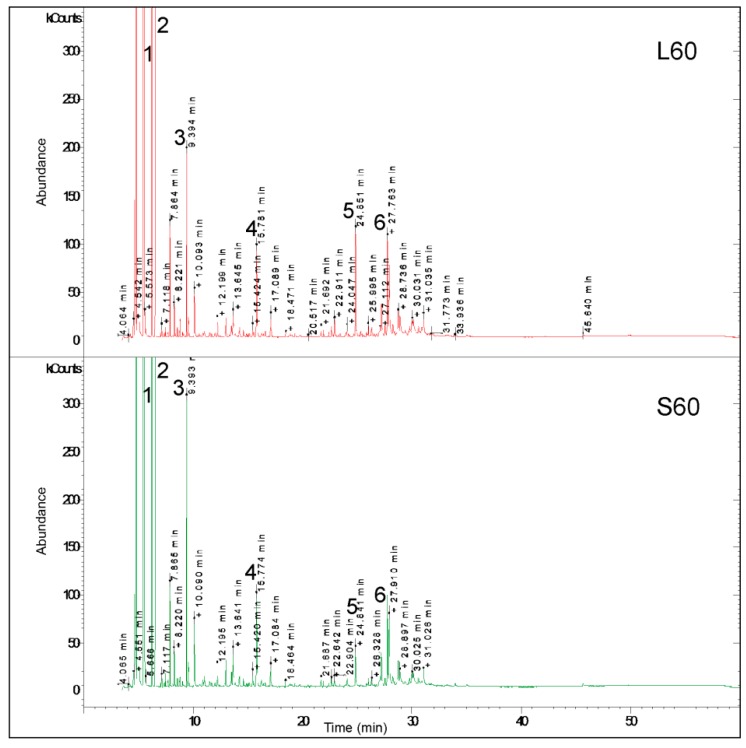
GC-MS chromatograms recorded for EO from *A. chamissonis* flower heads. L60—loamy soil at N dose 60 kg ha^−1^; S60—sandy soil at N dose 60 kg ha^−1^. Main EO components: 1—cumene; 2—alpha-pinene; 3—p-cymene; 4—decanal; 5—germacrene D; 6—spathulenol.

**Figure 3 molecules-24-04454-f003:**
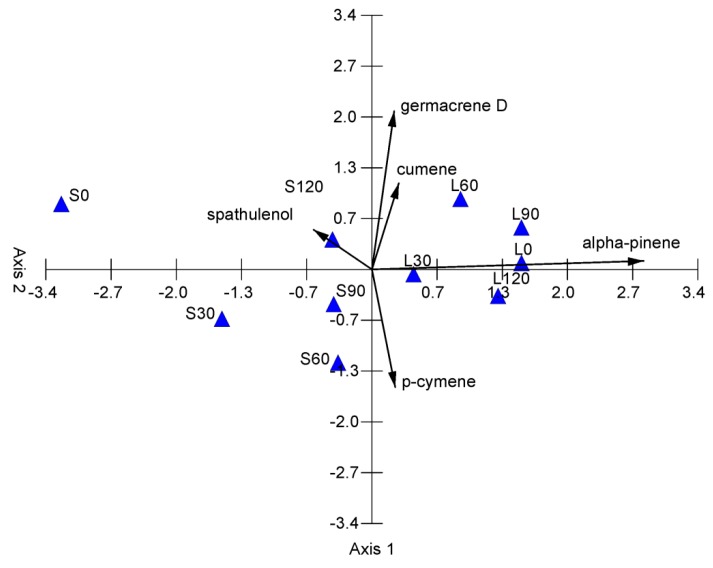
PCA diagram grouping the samples in terms of the EO composition of *A. chamissonis* flower heads depending on different nitrogen fertilization doses and soil types. S—sandy soil; L—loamy soil; 0—control, without N fertilization; 30—N dose of 30 kg ha^−1^; 60—N dose of 60 kg ha^−1^; 90—N dose of 90 kg ha^−1^; 120—N dose of 120 kg ha^−1^.

**Figure 4 molecules-24-04454-f004:**
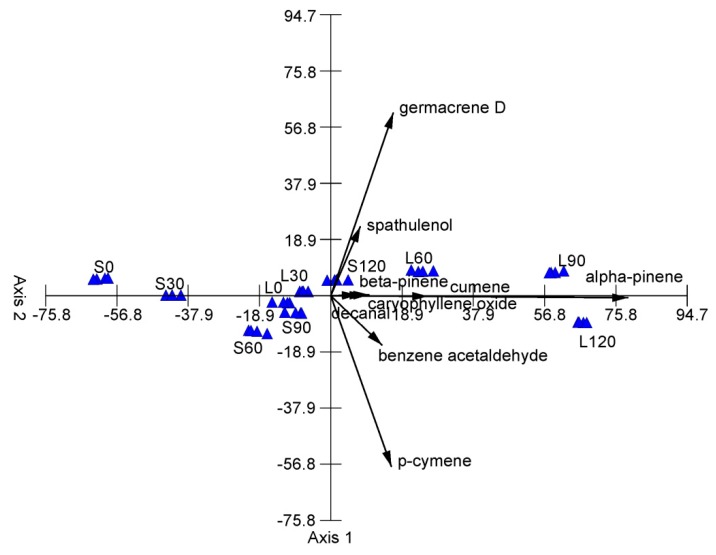
PCA diagram grouping the samples in terms of the yield of the main EO components of *A. chamissonis* flower heads depending on the different levels of nitrogen fertilization and soil type. Explanation: see Table 1.

**Table 1 molecules-24-04454-t001:** Crop yield, essential oil (EO) content and EO yield in *A. chamissonis* flower heads depending on the soil type and nitrogen (N) dose. L soil—loamy soil, S soil—sandy soil; 0, 30, 60, 90, 120—nitrogen doses (kg ha^−1^); the values designated by different letters are significantly different (*p* = 0.05) (Tukey’s test, *p* < 0.05).

Nitrogen Rate (kg ha^−1^)	0	30	60	90	120
Crop yield (kg ha^−1^)					
L soil	1210.0^b^ ± 38.3	1363.4^cd^ ± 26.2	1582.2^e^ ± 34.7	1837.2^f^ ± 26.6	1941.2^g^ ± 22.0
S soil	994.6^a^ ± 41.7	1177.1^b^ ± 31.1	1297.1^c^ ± 42.3	1430.0^d^ ± 30.7	1563.1^e^ ± 42.4
Two-way ANOVA results: soil (F = 793.19, *p* < 0.05), N (F = 508.24, *p* < 0.05), soil × N (F = 17.27, *p* < 0.05)					
EO content [%(v/w)]					
L soil	0.151^c^ ± 0.001	0.157^c^ ± 0.001	0.169^f^ ± 0.001	0.178^g^ ± 0.001	0.180^g^ ± 0.001
S soil	0.137^a^ ± 0.001	0.146^b^ ± 0.001	0.156^d^ ± 0.001	0.161^e^ ± 0.001	0.162^e^ ± 0.001
Two-way ANOVA results: soil (F = 1899.1, *p* < 0.05), N (F = 927.3, *p* < 0.05), soil × N (F = 10.8, *p* < 0.05)					
EO yield (g ha^−1^)					
L soil	1829.6^b^ ± 58.0	2144.7^c^ ± 40.8	267868^f^ ± 58.7	3264.7^g^ ± 47.4	3488.4^h^ ± 41.5
S soil	1354.9^a^ ± 46.8	1713.0^b^ ± 45.2	2022.1^c^ ± 65.9	2301.5^d^ ± 49.4	2529.1^e^ ± 58.5
Two-way ANOVA results: soil (F = 1820.1, *p* < 0.05), N (F = 1027.5, *p* < 0.05), soil × N (F = 48.9, *p* < 0.05)					

**Table 2 molecules-24-04454-t002:** Essential oil composition in *A. chamissonis* flower heads under the nitrogen impact. Explanation: see Table 1.

Compounds	RI	RI_Lit_	L0	L30	L60	L90	L120	S0	S30	S60	S90	S120
cumene	928	931	11.28	8.61	11.71	9.15	10.52	10.86	6.28	9.52	10.15	9.99
alpha-pinene	940	939	26.49	24.44	24.88	27.29	26.01	14.21	18.96	21.68	21.21	21.42
camphene	955	954	0.13	0.12	0.13	0.14	0.13	0.02	0.09	0.13	0.12	0.11
thuja-2.4(10)-diene	961	960	0.70	0.50	0.70	0.81	0.62	0.26	0.37	0.51	0.48	0.52
benzaldehyde	963	960	0.49	0.34	0.45	0.57	0.46	0.66	0.42	0.49	0.54	0.4
sabinene	978	975	0.20	0.17	0.16	0.18	0.18		0.16	0.19	0.19	0.16
beta-pinene	983	979	3.62	3.45	3.55	3.14	3.51	2.22	2.79	3.32	3.11	3.09
6-methyl-5-hepten-2-one	986	986	0.13	0.11	0.12	0.17	0.14		0.12	0.11	0.14	0.12
myrcene	996	991	1.62	1.62	1.68	1.36	1.76	1.44	1.62	1.81	1.77	1.60
mesitylene	999	996	0.10	0.08	0.11	0.12	0.09		0.06	0.08	0.09	0.09
n-octanal	1002	999	0.23	0.19	0.24	0.31	0.21		0.21	0.28	0.24	0.24
alpha-phellandrene	1012	1003	0.78	0.63	0.64	0.58	0.36	0.75	0.54	0.35	0.44	0.44
alpha-terpinene	1016	1017	0.13	0.13	0.15	0.12	0.12		0.09	0.11	0.11	0.13
beta-phellandrene (β-phellandrene)	1028	1030	0.33	0.31	0.38	0.31	0.29		0.29	0.25	0.28	0.32
p-cymene	1029	1025	7.40	5.99	5.47	5.69	7.85	5.15	6.40	9.06	7.13	5.94
limonene	1032	1029	0.70	0.67	0.61	0.53	0.77	0.53	0.64	0.83	0.75	0.61
benzene acetaldehyde	1041	1042	3.43	2.22	2.90	3.59	4.23	0.34	2.17	2.79	2.82	2.49
gamma-terpinene	1053	1060	0.07	0.11	0.10	0.06	0.06	0.02	0.06	0.06	0.08	0.06
alpha-methyl-benzene methanol	1056	1063	0.25	0.07	0.23	0.34	0.02	0.68	0.12	0.02	0.12	0.14
otrho-tolualdehyde	1062	1068	0.45	0.20	0.26	0.60	0.33	0.64	0.20	0.40	0.35	0.24
para-tolualdehyde	1064	1069	0.57	0.31	0.41	0.67	0.59	0.82	0.47	0.65	0.59	0.36
para-tolualdehyde	1077	1082	0.42	0.28	0.33	0.51	0.39		0.26	0.36	0.27	0.30
6-camphenone	1082	1097	0.20	0.18	0.21	0.24						0.17
linalool	1106	1097	0.28	0.23	0.29	0.28	0.26		0.28	0.26	0.25	0.22
nonanal	1110	1101	0.98	0.98	1.05	1.03	0.89	1.19	1.05	1.23	1.08	1.08
cis-para-menth-2-en-1-ol	1130	1122	0.27	0.18	0.18	0.26	0.20		0.20	0.21	0.20	0.17
alpha-campholenal	1135	1126	0.84	0.75	0.76	1.00	0.93	0.80	0.89	1.04	0.93	0.69
trans-pinocarveol	1147	1139	0.55	0.47	0.50	0.69	0.61	0.49	0.49	0.62	0.52	0.43
cis-verbenol	1150	1141	0.77	0.43	0.65	0.79	0.62	0.51	0.48	0.63	0.52	0.53
trans-verbenol	1154	1145	1.64	1.21	1.29	0.95	1.60	1.32	1.29	2.16	1.69	1.26
pinocarvone	1160	1165	0.53	0.43	0.53	0.67	0.53	0.59	0.46	0.52	0.58	0.55
para-mentha-1.5-dien-8-ol	1171	1170	0.42	0.29	0.40	0.39	0.40	0.34	0.36	0.39	0.33	0.24
terpinen-4-ol	1188	1177	0.20	0.13	0.17	0.18	0.15	0.18	0.14	0.15	0.16	0.11
naphthalene	1185	1181	0.13	0.13	0.16	0.17	0.12	0.13	0.12	0.13	0.15	0.12
myrtenal	1207	1196	0.79	0.69	0.72	0.80	0.87	0.85	0.73	0.86	0.69	0.63
safranal	1202	1197	0.28	0.24	0.29	0.35	0.29	0.45	0.26	0.30	0.32	0.26
decanal	1208	1202	4.48	4.52	4.79	4.37	3.76	6.20	4.74	4.94	4.48	4.45
trans-carveol	1223	1217	0.29	0.26	0.28	0.33	0.36	0.25	0.30	0.39	0.36	0.27
thymol methyl ether	1234	1235	0.13	0.14	0.17	0.16	0.28	0.00	0.18	0.19	0.20	0.16
carvacrol methyl ether	1249	1245	1.44	1.34	1.52	1.25	1.47	1.02	1.31	1.40	1.28	1.39
bornyl acetate	1288	1289	0.24	0.25	0.24	0.25	0.28		0.25	0.28	0.25	0.26
thymol	1304	1290	0.22	0.25	0.30		0.07				0.07	0.21
carvacrol	1313	1299		0.20	0.17				0.23	0.25	0.25	0.22
myrtenyl acetate	1327	1327		0.14								0.27
2E.4E-decadienal	1327	1317							0.2	0.31	0.20	
7-epi-silphiperfol-5-ene	1349	1348		0.12					0.13	0.08	0.14	0.10
eugenol	1383	1359	0.18	0.10	0.32	0.22	0.29		0.06	0.09	0.09	0.18
beta-maaliene	1388	1382	0.35	0.59	0.41	0.39	0.41	0.98	0.69	0.52	0.56	0.51
alpha-isocomene	1394	1388	0.26	0.39	0.30	0.28	0.25	0.82	0.44	0.29	0.36	0.38
cyperene	1405	1399	0.13	0.19	0.15	0.14	0.09	0.48	0.19	0.15	0.16	0.18
E-caryophyllene	1417	1419	0.43	0.68	0.55	0.22	0.37		0.85	0.68	0.71	0.64
beta-duprezianene< >	1424	1423	0.61	0.87	0.74	0.63	0.45	1.58	0.81	0.52	0.68	0.85
(E)-alpha-ionone	1428	1430		0.16	0.14		0.13			0.19	0.21	0.19
beta-copaene	1435	1432		0.07	0.02				0.07		0.07	0.09
(Z)-beta-farnesene	1463	1443	0.10	0.22	0.17	0.13	0.17		0.21	0.16	0.19	0.26
alpha-humulene	1465	1455	0.33	0.53	0.43	0.36	0.29	0.91	0.51	0.33	0.44	0.53
gamma-muurolene	1487	1480		0.15	0.12	0.09						0.16
germacrene D	1494	1485	4.98	4.74	6.86	6.39	4.65	5.42	3.96	1.93	3.29	6.07
(Z)-alpha-bisabolene	1498	1507		0.09	0.06		0.29		0.30	0.38	0.24	0.05
alpha-bulnesene	1509	1510	0.07			0.18		0.46	0.19			0.23
delta-amorphene	1512	1512						0.27	0.10			0.10
gamma-cadinene	1525	1514	0.15	0.23	0.06	0.08	0.14	0.49	0.24	0.15	0.14	0.16
delta-cadinene	1529	1523	0.44	0.74	0.62	0.40	0.43	1.06	0.64	0.45	0.56	0.72
10-epi-cubebol	1546	1535					0.10		0.15	0.14	0.18	
lippifoli-1(6)-en-5-one	1560	1553	1.34	2.03	2.03	2.05	1.83	3.08	2.26	2.00	2.07	2.11
spathulenol	1582	1578	4.05	5.01	4.54	5.48	4.02	7.36	6.24	4.49	4.52	5.10
caryophyllene oxide	1587	1583	3.98	4.90	4.55	5.17	4.45	5.64	5.10	4.99	4.93	5.01
salvial-4(14)-en-1-one	1604	1595	0.62	0.92	0.81	0.64	0.65		1.11		0.97	1.08
humulene epoxide II	1623	1608	2.36	2.07	1.60	1.39	1.62	2.47	2.65	1.99	2.16	2.16
epi-alpha-cadinol	1655	1640	1.01	2.09	1.25	1.25	1.36	2.77	2.91	1.74	2.08	2.40
epoxy allo-alloaromadendrene	1661	1641	0.33	0.66	0.36	0.38	0.40	0.76	0.80	0.44	0.53	0.79
14-hydroxy-9-epi-(E)-caryophyllene	1669	1670	1.78	1.41	1.13	1.03	1.06	2.84	2.06	1.22	1.30	1.41
valeranone	1692	1685	0.60	1.00	0.78	0.66	0.94	2.48	1.33	1.06	1.06	1.19
guaia-3.10(14)-dien-11-ol	1704	1678	0.42	0.71	0.54	0.49	0.30	1.68	0.95	0.41	0.61	0.83
khusinol	1708	1680	1.08	1.67	1.36	1.51	1.13	3.01	2.68	1.31	1.52	1.80
Monoterpene Hydrocarbons			53.74	47.01	50.44	49.69	52.54	35.71	38.59	48.21	46.18	44.66
Aromatic Aldehydes			5.36	3.35	4.35	5.94	6.00	2.46	3.52	4.69	4.57	3.79
Aliphatic Aldehydes			5..69	5.69	6.08	5.71	4.86	7.39	6.00	6.45	5.80	5.77
Sesquiterpene Hydrocarbons			11.30	13.87	14.15	12.75	10.96	19.15	14.45	9.49	11.65	15.53
Sesquiterpene Alcohols			6.56	9.48	7.69	8.73	6.91	14.82	12.93	8.09	8.91	10.13
Oxygenated Sesquiterpenes			3.98	4.90	4.55	5.17	4.45	5.64	5.10	4.99	4.93	5.01
Others			13.17	12.03	12.52	11.97	12.43	12.31	13.32	13.05	13.22	12.85
Sum of Identified (%)			99.80	96.33	99.78	99.96	98.15	97.48	93.91	94.97	95.26	97.74

RI—retention indices (from temperature-programming using definition of Van Den Dool and Kratz [61]). RI_Lit_—retention indices taken from literature [62].

**Table 3 molecules-24-04454-t003:** Results of PCA based on essential oil ingredients. (**a**) Eigenvalues and variance (%) explained by the first two PCA axes; (**b**) Loading components for each variable associated with the two axes.

Chemical Variables	Axis 1	Axis 2
(**a**)		
Eigenvalues	20.798	4.557
Percentage	70.317	15.407
Cumulative percentage	70.317	85.724
(**b**)		
alpha-pinene	0.876	0.035
germacrene D	0.072	0.655
p-cymene	0.075	−0.487
cumene	0.087	0.357
spathulenol	−0.188	0.164

**Table 4 molecules-24-04454-t004:** Yield of essential oil from *A. chamissonis* flower heads depending on the soil type and nitrogen (N) fertilization. Explanation: see Table 1.

Nitrogen rate (kg ha^−1^)	Soil	0	30	60	90	120
alpha-pinene	L	482.65^d^ ± 15.35	524.16^e^ ± 5.08	665.38^f^ ± 14.61	894.17^g^ ± 12.93	905.17 ^g^ ± 5.60
	S	196.82^a^ ± 8.12	325.01^b^ ± 8.58	438.40^c^ ± 14.29	488.15^d^ ± 10.49	542.30^e^ ± 14.68
Two-way ANOVA results: soil (F = 6583.8, *p* < 0.05), N (F = 1639.0, *p* < 0.05), soil × N (F = 112.6, *p* < 0.05)						
cumene	L	205.52^d^ ± 6.54	184.66^c^ ± 1.79	313.17^h^ ± 6.88	299.80^g^ ± 4.33	366.10^i^ ± 2.27
	S	149.37^b^ ± 6.16	107.65^a^ ± 2.84	192.51^c^ ± 6.28	233.61^e^ ± 5.02	252.92^f^ ± 6.85
Two-way ANOVA results: soil (F = 2780.5, *p* < 0.05), N (F = 1316.89, *p* < 0.05), soil × N (F = 59.4, *p* < 0.05)						
p-cymene	L	134.83^c^ ± 4.29	128.47^c^ ± 1.24	146.29^d^ ± 3.21	186.44^f^ ± 2.70	273.19^g^ ± 1.69
	S	70.83^a^ ± 2.92	109.71^b^ ± 2.90	183.20^f^ ± 5.97	164.10^e^ ± 3.52	150.39^d^ ± 4.07
Two-way ANOVA results: soil (F = 1222.2, *p* < 0.05), N (F = 1276.9, *p* < 0.05), soil × N (F = 586.4, *p* < 0.05)						
germacrene D	L	90.74^d^ ± 2.89	101.66^e^ ± 0.98	183.46^h^ ± 4.03	209.37^i^ ± 3.03	161.82^g^ ± 1.00
	S	74.55^bc^ ± 3.08	67.88^b^ ± 1.79	39.03^a^ ± 1.27	75.72^c^ ± 1.63	153.68^f^ ± 4.16
Two-way ANOVA results: soil (F = 6527.5, *p* < 0.05), N (F = 1307.7, *p* < 0.05), soil × N (F = 1237.7, *p* < 0.05)						
spathulenol	L	73.79^a^ ± 2.35	107.45^d^ ± 1.04	121.42^e^ ± 2.67	179.55^h^ ± 2.60	139.90^g^ ± 0.87
	S	101.23^c^ ± 4.18	106.97^d^ ± 2.82	90.79^b^ ± 2.96	104.03^cd^ ± 2.23	129.12^f^ ± 3.49
Two-way ANOVA results: soil (F = 465.4, *p* < 0.05), N (F = 554.6, *p* < 0.05), soil × N (F = 389.3, *p* < 0.05)						
decanal	L	81.63^a^ ± 2.60	96.94^b^ ± 0.94	128.10^e^ ± 2.81	143.19^f^ ± 2.07	130.85^e^ ± 0.81
	S	85.27^a^ ± 3.52	81.25^a^ ± 2.14	99.89^bc^ ± 3.26	103.11^c^ ± 2.21	112.66^d^ ± 3.05
Two-way ANOVA results: soil (F = 646.3, *p* < 0.05), N (F = 455.6, *p* < 0.05), soil × N (F = 77.0, *p* < 0.05)						
caryophyllene oxide	L	72.52 ^a^ ± 2.60	105.09 ^c^ ± 0.94	121.68 ^e^ ± 2.81	169.40 ^g^ ± 2.07	154.86^f^ ± 0.81
	S	77.57^a^ ± 3.20	87.42^b^ ± 2.31	100.90^c^ ± 3.29	113.47^d^ ± 2.44	126.84^e^ ± 3.43
Two-way ANOVA results: soil (F = 877.2, *p* < 0.05), N (F = 1032.3, *p* < 0.05), soil × N (F = 140.4, *p* < 0.05)						
beta-pinene	L	65.96^c^ ± 2.10	73.99^d^ ± 0.72	94.94^f^ ± 2.09	102.88^g^ ± 1.49	122.15^h^ ± 0.76
	S	30.53^a^ ± 1.26	47.83 ^b^ ± 1.26	67.13 ^c^ ± 2.19	71.58 ^d^ ± 1.54	78.23 ^e^ ± 2.12
Two-way ANOVA results: soil (F = 4080.3, *p* < 0.05), N (F = 1296.1, *p* < 0.05), soil × N (F = 39.8, *p* < 0.05)						
benzene acetaldehyde	L	62.49^e^ ± 1.99	47.61^c^ ± 0.46	77.56^f^ ± 1.70	117.63^g^ ± 1.70	147.21^h^ ± 0.91
	S	4.68^a^ ± 0.19	37.20^b^ ± 0.98	56.42^d^ ± 1.84	64.90^e^ ± 1.39	63.04^e^ ± 1.71
Two-way ANOVA results: soil (F = 10,265.3, *p* < 0.05), N (F = 3750.1, *p* < 0.05), soil × N (F = 886.7, *p* < 0.05)						

**Table 5 molecules-24-04454-t005:** Results of PCA based on essential oil yield. (**a**) Eigenvalues and variance (%) explained by the first two PCA axes; (**b**) Loading components for each variable associated with the two axes.

Chemical Variables	Axis 1	Axis 2
(**a**)		
Eigenvalues	59,339.4	1823.14
Percentage	94.520	2.904
Cumulative percentage	94.520	97.424
(**b**)		
alpha-pinene	0.888	−0.008
germacrene D	0.187	0.692
p-cymene	0.181	−0.647
spathulenol	0.088	0.261
benzene acetakdehyde	0.153	−0.187
cumene	0.290	−0.003
caryophyllene oxide	0.114	0.003
beta-pinene	0.103	0.003
decanal	0.080	0.002

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
