# Peer review of "Chemical Composition of Essential Oil from Flower Heads of Arnica Chamissonis Less. under a Nitrogen Impact"

_molecules, 2019, doi:10.3390/molecules24244454_

Round 1

Reviewer 1 Report

The manuscript presents interesting findings regarding Arnica chamissonis essential oils under nitrogen impact. It is good experimental article with interesting subject and good experimental work. The manuscript is well written and includes a great deal of  information, which is reflected in the significant number of  references listed. The methodology of experimental part is well established and it does not raise any objections. The results and discussion are represented in a logical way. Authors included 2 figures and 5 tables which are clear and legible. Below please find my comments.

Table 2 – please provide the names of groups of compounds and put their sum of content.

Which reference flora was used to identify the species?

Subsect. 4.3.1. Assay of the essential oil content. Please provide solid to liquid ratio. How many repeats of distillation were performed?

Author Response

Responses to the remarks of Reviewer 1

We would like to thank the reviewer for the valuable comments that helped us to significantly improve the manuscript. The detailed responses to the reviewer’s comments are given below.

Comments and Suggestions for Authors

The manuscript presents interesting findings regarding Arnica chamissonis essential oils under nitrogen impact. It is good experimental article with interesting subject and good experimental work. The manuscript is well written and includes a great deal of  information, which is reflected in the significant number of  references listed. The methodology of experimental part is well established and it does not raise any objections. The results and discussion are represented in a logical way. Authors included 2 figures and 5 tables which are clear and legible.

Below please find my comments.

Table 2 – please provide the names of groups of compounds and put their sum of content.

Table 2 has been corrected and the names of groups of compounds and the sum of their content have been included according to the suggestions of the reviewer.

Which reference flora was used to identify the species?

Plant species was identified by Dr. Anna Rysiak, a taxonomist from the University of Maria Curie-Skłodowska in Lublin based on the reference material from the collection of medicinal plants of the Department of Industrial and Medicinal Plants of the University of Life Sciences in Lublin. This information has been added in the Materials and Methods section (Par. 4.1).

Subsect. 4.3.1. Assay of the essential oil content. Please provide solid to liquid ratio. How many repeats of distillation were performed?

The lacking information has been added in the text of the revised manuscript as suggested by the reviewer (Par. 4.3.1). The solid to liquid ratio was 20 g of powdered plant material to 500 mL water. The single distillation (according to the Polish Paharmacopoeia VI) was performed in four repetitions.

Additionally, the names of chemical compounds have been unified and figures 1 and 2 have been corrected according to the remarks of the reviewer. The style and grammar have also been improved within the text of the whole manuscript.

Reviewer 2 Report

Article is interesting and the study was well conducted, but must be corrected before possible publication. Methodological errors and text editing are required:

1) Line 162: Insert RI from literature used for identifying the compounds.

2) Draw chemical structures of the main compounds of the essential oils.

3) Insert overlapping chromatograms of the essential oils with L soil and others containing S soil.

4) Explain how the compounds were quantified. Were calibration curves performed for the reported compounds?

5) Format the characters highlighted in yellow, as shown in the attachment.

6) Set the references according to the journal's standards.

7) Discuss variations in chemical composition based on some biochemical aspects.

8) Insert conclusion.

Author Response

Responses to the remarks of Reviewer 2

We would like to thank the reviewer for the valuable comments that helped us to significantly improve the manuscript. The detailed responses to the reviewer’s comments are given below.

Article is interesting and the study was well conducted, but must be corrected before possible publication.

Methodological errors and text editing are required:

1) Line 162: Insert RI from literature used for identifying the compounds.

Table 2 has been corrected as suggested. RI from literature used for identifying the compounds has been included.

2) Draw chemical structures of the main compounds of the essential oils.

Chemical structures of the main compounds of the essential oils have been drawn and presented in the added figure (Figure 1 in the revised version of the manuscript). The other figures have been reordered and renumbered.

3) Insert overlapping chromatograms of the essential oils with L soil and others containing S soil.

The overlapping chromatograms of the essential oils from A. chamissonis flower heads obtained from L and S soils have been presented in the added figure (Figure 2 in the revised version of the manuscript). Consequently, all figures have been reordered and renumbered.

4) Explain how the compounds were quantified. Were calibration curves performed for the reported compounds?

The quantitative analysis of the identified compounds was performed based on calibration curves and additionally checked/confirmed using internal standards (alkanes C12 and C19) as described by Kowalski and Wawrzykowski 2009 [95]. In details, as described in this publication, the quantitative analysis was performed on the basis of calibration curves plotted to find the dependence between the ratio of peak area for the analyte to the area for internal standard (Aanalyte:Ai.s.) vs. the analyte concentration (Canalyte), for p-cymene, γ-terpinene, linalool, thymol, carvacrol, (E)-caryophyllene, caryophyllene oxide, in appropriate concentration range. The following alkanes were applied as internal standards: C12 (for compounds with retention index <1300, p-cymene, γ-terpinene, linalool); and C19 (for compounds with retention index >1300, thymol, carvacrol, (E)-caryophyllene, caryophyllene oxide). The contents of the analysed substances were read from achieved calibration curves, the data for which originated from peak areas for thyme oil components and internal standard peak areas from GC separation. The final result took into account all dilutions during the whole analytical procedure. For comparative purposes, the percentage of main components of the arnica essential oil was presented, assuming that the sum of peak areas for all identified constituents was 100%.

5) Format the characters highlighted in yellow, as shown in the attachment.

Corrected in accordance with the recommendations of the reviewer.

Additionally, the names of chemical compounds have been unified and figures 1 and 2 have been corrected according to the remarks of the reviewer. The style and grammar have also been improved within the text of the whole manuscript.

6) Set the references according to the journal's standards.

Corrected as recommended by the reviewer.

7) Discuss variations in chemical composition based on some biochemical aspects.

The discussion has been supplemented and improved according to the suggestion of the reviewer.

8) Insert conclusion.

Conclusion has been included in the revised version of the manuscript.

Round 2

Reviewer 2 Report

Authors performed all corrections, but they must correct the legend of Figure 2 before publication. Thus, change "HPLC" to "GC-MS".

Author Response

Dear Reviewer,

The legend of Figure 2 was corrected.

We would like to thank the reviewer for the valuable comments that helped us to significantly improve the manuscript.

Sincerely yours,
Małgorzata Wójcik
